

# EIF4A3 serves as a prognostic and immunosuppressive microenvironment factor and inhibits cell apoptosis in bladder cancer

Bing Hu[*], Ru Chen[*], Ming Jiang, Situ Xiong, Xiaoqiang Liu and Bin Fu

Department of Urology, The First Affiliated Hospital of Nanchang University, Nanchang, China
Jiangxi Institute of Urology, Nanchang, China
[*] These authors contributed equally to this work.

## ABSTRACT

EIF4A3 (Eukaryotic translation initiation factor 4A3 (EIF4A3) was recently recognized as an oncogene; however, its role in BLCA (bladder cancer) remains unclear. We explored EIF4A3 expression and its prognostic value in BLCA in public datasets, including the TCGA (The Cancer Genome Atlas) and GEO (Gene Expression Omnibus). Thereafter, the association between EIF4A3 expression and the infiltration of immune cells and immune-checkpoint expression was determined using TIMER2 (Tumor Immune Estimation Resource 2) tool. Additionally, the impact of EIF4A3 on cellular proliferation and apoptosis events in BLCA cell lines was determined by siRNA technology. In this study, EIF4A3 was found to be significantly upregulated in BLCA, upregulated expression of EIF4A3 was related to poor prognosis, advanced histologic grade, subtype, pathological stage, white race, and poor primary therapy outcome. The immune infiltration analysis revealed that EIF4A3 expression was negatively associated with CD8$^+$ and CD4$^+$ T cells and positively with myeloid-derived suppressor cells, macrophage M2, cancer-associated fibroblasts, and Treg cells. Moreover, EIF4A3 was coexpressed with PD-L1 (programmed cell death 1-ligand 1) and its expression was higher in patients responding to anti-PD-L1 therapy. EIF4A3 knockdown significantly inhibited proliferation and promoted apoptosis in 5,637 and T24 cells. In summary, BLCA patients with elevated EIF4A3 expression had an unfavorable prognosis and immunosuppressive microenvironment, and EIF4A3 may facilitate BLCA progression by promoting cell proliferation and inhibiting apoptosis. Furthermore, our study suggests that EIF4A3 is a potential biomarker and therapeutic target for BLCA.

# INTRODUCTION

Bladder cancer (BLCA) is the second most common malignant tumor of the genitourinary system after prostate cancer, ranking ninth in the global incidence of cancers and sixth in men. According to the most recent data, it is projected that in 2023 there will be 82,290 newly diagnosed cases of bladder cancer and 16,710 related deaths in the United

Corresponding authors
Xiaoqiang Liu, shaw177@163.com
Bin Fu, urofubin@126.com

States (*Bray et al., 2018*; *Siegel et al., 2023*). Approximately 70–80% of patients present with bladder cancer in the non-muscle invasive stage at the initial diagnosis, but up to 20% of patients progress to high-grade, highly staged muscle-invasive bladder cancer with a 5-year survival rate of less than 50% (*Kohada et al., 2021*; *Wallerand et al., 2011*). Although many advancements have been made in BLCA treatment, including target therapy and immunotherapy, there has been no notable increase in the survival rate for BLCA over the past three decades (*Berdik, 2017*; *Ghasemzadeh et al., 2016*; *Pond et al., 2021*). Therefore, investigating the underlying mechanism of BLCA development and identifying a new therapeutic target is crucial to improve the patient's prognosis.

EIF4A3 (eukaryotic translation initiation factor 4A3), also known as DDX48, is a DEAD-box protein that catalyzes translation by releasing sub-structures from the 5′-untranslated region of the mRNA molecules (*Linder & Jankowsky, 2011*). EIF4A3 primarily localizes to the nucleus and is a core unit of EJC (exon-exon junction complex) which regulates mRNA splicing, translation, localization, and NMD (nonsense-mediated decay) (*Ferraiuolo et al., 2004*; *Palacios et al., 2004*). Several studies reported that EIF4A3 could upregulate circRNA expression by binding to the corresponding mRNA transcripts, thus inducing carcinogenesis and development (*Wang et al., 2018*; *Zheng et al., 2020*). Additionally, it acts as an RNA-binding protein, which could stabilize long non-coding RNAs and mRNAs and promote tumor development (*Li et al., 2019*; *Tang et al., 2020*; *Yang et al., 2020*). For example, EIF4A3 knockdown attenuated glioblastoma cell migration, invasion, and proliferation (*Tang et al., 2020*). Recently, *Wei et al. (2021)* reported that EIF4A3 could upregulate circ0008399 expression resulting in impaired sensitivity to chemotherapy in BLCA. Similarly, EIF4A3 was also recruited by circSEMA5A to improve SEMA5A stability, resulting in mRNA overexpression, thereby facilitating BLCA progression through the miR-330-5p/ENO1 signaling pathway (*Wang et al., 2020*). Although these studies associated EIF4A3 expression with BLCA, the direct effect of EIF4A3 on established BLCA has not been determined. An exploration of how EIF4A3 contributes to bladder carcinogenesis may enable its use as a prognostic factor or even as a therapeutic target.

The purpose of the current study was to investigate the role of EIF4A3 in predicting the prognosis, correlation with immune infiltration, and potential mechanism in the development of tumorigenesis in BLCA. Therefore we compared the EIF4A3 expression between cancer and adjacent tissues in TCGA (The Cancer Genome Atlas) dataset and explored its prognostic value and further validated it in the external GEO (Gene Expression Omnibus) (GSE32894, GSE32548, GSE31684, and GSE13507) datasets. Simultaneously, the relationship between EIF4A3 expression and immune infiltration, and immune checkpoint gene expression was analyzed using the TIMER2 (Tumor Immune Estimation Resource version 2.0) tool. The signaling pathways potentially regulated by EIF4A3 overexpression were analyzed by GSEA (gene set enrichment analysis). Finally, we observed the function of EIF4A3 on BLCA cell proliferation and apoptosis *in vitro*.

## MATERIAL AND METHOD

### Gene expression profile

The pan-cancer expression of EIF4A3 was obtained from TIMER2 (http://timer.cistrome.org/) (*Li et al., 2020*), and the TCGA database was used to collect mRNA expression data and clinical information of BLCA. The differential expression of EIF4A3 in BLCA patients exhibiting different clinical features was analyzed using the R statistical environment (version 3.6.3).

### Survival analysis

The GSE32894, GSE32548, and GSE31684 datasets, including the mRNA expression data and clinical records, were acquired from the GEO (https://www.ncbi.nlm.nih.gov/geo/) database. The R packages of "survival" and "survminer" were used to analyze and visualize the OS(overall survival) difference in the three GEO datasets and different sub-groups based on the TCGA dataset. Additionally, PrognoScan (http://dna00.bio.kyutech.ac.jp/PrognoScan/index.html) (*Mizuno et al., 2009*) tool to was applied to perform the OS and DSS (disease special survival) analysis of the patients in the GSE13507 dataset. Furthermore, the Kaplan–Meier plotter (http://kmplot.com/analysis/) tool was used to analyze the OS difference in patients with enriched and decreased $CD8^+$ T cells, based on the TCGA dataset.

### A nomogram constructed to predict BLCA survival and its evaluation

Based on the TCGA cohort, EIF4A3 expression and all statistically significant clinical factors in univariate analysis were included to construct the nomogram using the "rms" package. The "timeROC" package was utilized to get the 1, 3, and 5-year curves. The calibration curve was also performed *via* "rms" package.

### Immune infiltration, gene expression correlation analysis, and anti-PD-L1 therapy response prediction

The correlation between EIF4A3 expression and the infiltration of different immune cells, including $CD8^+$ T cells, $CD4^+$ T cells, B cells, NK (natural killer) cells, macrophage cells, MDSCs (myeloid-derived suppressor cells), CAFs (cancer-associated fibroblasts), macrophage M2, and Treg (regulatory T) cells, was determined using the TIMER2 tool, Tumor purity was applied to make adjustments in immune infiltration analysis.

The correlation analysis between EIF4A3 expression and the immune checkpoint, including PD1 (programmed cell death 1, also known as PDCD1), PD-L1 (programmed cell death 1-ligand 1, also known as CD274), and CTLA4 (CTL-associated antigen 4), was also explored using the Gene_Corr module on TIMER2 (with tumor purity adjustment). Two external GEO (GSE32548, GSE32894) datasets were used to validate the correlation between EIF4A3 and PD-L1 expression using Spearman's rank correlation coefficient, and the results were visualized *via* "ggplot2" packages.

Furthermore, we downloaded the "IMvigor210CoreBiologies" (http://research-pub.gene.com/IMvigor210CoreBiologies/packageVersions/) package containing anti-PD-L1 therapy clinical trial outcome for uroepithelial carcinoma (*Mariathasan et al., 2018*).

According to the therapy response, the differences in the EIF4A3 expression of the patients were compared and the "ggplot2" package was used to visualize the results.

## Gene set enrichment analysis

GSEA (gene set enrichment analysis) (*Subramanian et al., 2005*) was performed between high- and low-EIF4A3 expression groups using gene set from MSigDB Collections and the potential signaling pathways that EIF4A3 regulates were identified using the "clusterProfiler" package.

## Cell culture and transfection

The human uroepithelial cancer cell lines T24, 5,637, and UM-UC-3, as well as human normal uroepithelial cell line SV-HUC-1, were bought from the cell bank of Type Culture Collection of Chinese Academy of Science, Shanghai Institute of Cell Biology (Shanghai, China). T24, 5,637, UM-UC-3, and SV-HUC-1 cells were cultured in DMEM, MEM, RPIM-1640, and F12K media, respectively, supplemented with 10% fetal bovine serum (FBS, Hyclone) and 100 U/mL penicillin/streptomycin and grown at 37 °C, 5% $CO_2$. The T24 and 5,637 cells were seeded into six-well plates ($2 \times 10^5$ cells/well), and small interference RNA (siRNA) and control siRNA (si-NC) were transferred to the cells using lipofectamine 2000 (Thermo Fisher Scientific, Waltham, MA, USA), as per the manufacturer's instructions The siRNA sequence of EIF4A3(RiboBio, Guangzhou, China) was as follows: siRNA-1: 5′-CGAGCAATCAAGCAGATCA-3′, siRNA-2: 5′-GCTGGATTACGGACAGCAT-3′.

## RNA extraction and real-time quantitative RT-PCR (qRT-PCR) analysis

The TRIzol reagent (Invitrogen, Waltham, MA, USA) was used to extract the total RNA of the clinical specimen and transfected cells. All specimens were obtained with the patient's consent and signed an informed consent form. The study has been approved by the Ethics Committee of the First Affiliated Hospital of Nanchang University. cDNA was synthesized using the FastKing RT Kit (with gDNase; TIANGEN Biotech, Beijing, China), and PCR was performed in the ABI PRISM 7500 real-time PCR machine (Applied Biosystems, Waltham, MA, USA) following the manufacturer's instructions and cycling conditions were as follows: denaturation at 95 °C for 1 min, amplification at 95 °C for 15s and 60 °C for 30s for 40 cycles. The primer sequences used in this analysis were as follows: EIF4A3: forward primer: 5′-CAACGAGCAATCAAGCAG-3′ and reverse primer: 5′-GTGGGAGCCAAGATCAAA-3′; ACTB: forward primer: 5′-TCTCCCAAGTCCACACAGG-3′ and reverse primer: 5′-GGCACGAAGGCTCATCA-3′.

## Western blot

The transfected cells and tissue lysates were prepared, fractionated using sodium dodecyl sulfate-polyacrylamide gel electrophoresis, and followed transferred to polyvinylidene fluoride (PVDF) membranes. Subsequently, the membranes were blotted in 5% nonfat milk for 1 h and incubated overnight at 4 °C with the following primary antibodies: GAPDH (ab8245, 1:1000; Abcam, Cambridge, UK), EIF4A3 (sc-365549, 1:1000), PARP (9532, 1:1000; CST, Danvers, MA, USA), Caspase3 (9662, 1:1000; CST, Danvers, MA,

USA), Bcl-2 (15071, 1:1000; CST, Danvers, MA, USA). Thereafter, the membranes were washed three times with tris-buffered saline with 0.1% Tween 20 and exposed to secondary antibodies (anti-mouse for GAPDH, EIF4A3, and anti-rabbit for PARP, Caspase3, Bcl-2) for 1 h at room temperature. Lastly, the chemiluminescence detection reagent (WBKLS0100, Millipore, Burlington, MA, USA) was applied to scan and visualize the protein bands.

## Cell proliferation and cloning assay

The effects of EIF4A3 on the vitality of BLCA cells were assessed using the CCK-8 detection kit (Keygen, Nanjing, China). The si-RNA-treated T24 and 5,637 cells were seeded into a 96-well plate, (3,000 cells/well) after 24 h, 48 h, 72 h, and 96 h, of incubation, the cells were incubated with the CCK-8 solution for 2 h at 37 °C and their absorbance were measured at 450 nm. The cells were also seeded in a 6-well plate (400 cells/well) to detect their proliferation ability, after 7-14 d of incubation, the cells were fixed and stained by 6% glutaraldehyde and 0.5% crystal violet. Image J was used to count clone numbers.

## Cell apoptosis analysis

To investigate cell apoptosis events, the Annexin V-FITC/PI Cell Apoptosis Detection Kit (Transgen, Beijing, China) was used. Firstly, cells were digested with EDTA-free trypsin, washed twice with cold PBS, and then resuspended with binding buffer. Subsequently, cells were incubated in the dark with 5 μl Annexin V-FITC and/or 5 μl PI for 15 min. The detection of apoptosis events was carried out using a flow cytometer (BriCyte E6, Mindray, Shenzen, China) and analyzed using the FlowJo software.

## Statistical analysis

R (version 3.6.3) and GraphPad Prism 7.01 were used for statistical analysis. The association between EIF4A3 expression and clinical characteristics was analyzed by the Wilcoxon rank-sum test, Kruskal-Wallis test, and Fisher's exact test. The survival rate of the patients was determined by the Kaplan–Meier method. The other continuous data were analyzed by student's $t$-test. Univariate and multivariate Cox regression analysis was used to screen out the independent prognosis factors. $P < 0.05$ was considered statistically significant.

## RESULTS

In this study, we found that EIF4A3 is upregulated in BLCA, and is related to the advanced histologic grade, subtype, pathological stage, race, poor primary therapy outcome, and low OS and DSS. Overexpressed EIF4A3 was considered an independent prognosis biomarker in patients with BLCA. All the above results reveal that EIF4A3 could be a novel prognosis factor for BLCA. Additionally, infiltration of immunosuppressive cells and expression of PD-L1 were positively correlated with the expression of EIF4A3 which was also elevated in patients who responded to anti-PD-L1 immunotherapy. The *in vitro* results revealed that knockdown of EIF4A3 substantially inhibited proliferation and increased apoptosis in 5,637 and T24 cells.

### EIF4A3 expression is upregulated in BLCA

A TIMER2 analysis of EIF4A3 expression in tumors and normal tissues indicated that cancer expressed EIF4A3 notably more than normal tissues, such as BLCA, breast cancer,

Cholangiocarcinoma, Colon adenocarcinoma, Esophageal carcinoma, etc. compared to the normal tissues. In contrast, EIF4A3 expression was lower in kidney chromophobe, kidney renal clear cell carcinoma, and thyroid carcinoma compared to the normal tissues (Fig. 1A). Furthermore, combined analysis of TCGA and GTEx (the genotype-tissue expression) data revealed that EIF4A3 expression was significantly upregulated in BLCA ($p < 0.001$) (Fig. 1B). Moreover, paired bladder cancer samples (from the same patient) mRNA expression data from TCGA also revealed a significantly high EIF4A3 expression in tumor tissues ($p < 0.001$) (Fig. 1C). Subsequently, a correlation between clinical characteristics and EIF4A3 expression levels was determined by grouping patients based on their clinical characteristics. These results showed that the higher expression of EIF4A3 was significantly correlated with histologic grade ($p < 0.001$), pathological subtype ($p < 0.001$), pathologic stage ($p < 0.05$), race ($p < 0.01$), primary therapy outcome ($p < 0.01$), OS ($p < 0.01$), and DSS ($p < 0.05$) (Figs. 1D–1J). A median cut-off was considered, according to the EIF4A3 expression, and differential analysis was performed between the low and high EIF4A3 expression groups based on different clinical features (Table 1).

Furthermore, qRT-PCR assay using BLCA cell lines and 22 paired BLCA samples, to verify the mRNA expression levels of EIF4A3 in BLCA, revealed that EIF4A3 expression was significantly elevated ($p < 0.01$) in 5,637, T24, and UM-UC-3 cells compared to the SV-HUC-1 cell line (Fig. 2A). Similarly, a significant increase in EIF4A3 mRNA expression was found in cancer tissues compared with paracancerous tissues in the BLCA tissue samples ($p < 0.01$; Fig. 2B). And western blot results also revealed that EIF4A3 expression was increased in both cancer cell lines and tumor tissues (Figs. 2C–2D). These results reveal that EIF4A3 expression was upregulated in BLCA at both transcription and translation levels.

## EIF4A3 is an independent prognostic factor in BLCA and validated in the external databases

To identify the prognostic value of EIF4A3 in BLCA, the patients in the TCGA dataset were divided into low and high EIF4A3 expression groups, based on the median EIF4A3 expression, to perform survival analysis. As shown in Figs. 3A–3B, high EIF4A3 expression was related to poor OS (HR =1.57, $p = 0.003$) and DSS (HR =1.56, $p = 0.017$) in BLCA. To further verify the prognostic value of EIF4A3 expression, we screened four external independent GEO datasets. Among these, GSE32894, GSE32548, and GSE31684 showed a lower OS for BLCA patients with higher EIF4A3 expression (HR =4.72, $p < 0.001$, HR =2.72, $p = 0.017$, HR =1.77, $p = 0.033$ respectively) (Figs. 3C–3E). Likewise, Figs. 3F–3G suggested that high expression group patients had a worse OS (HR =1.59, $p = 0.016$) and DSS (HR =2.70, $p = 0.0002$) in the GSE13507 cohort, which was obtained from the PrognoScan database.

In addition, the subgroup analysis showed that high EIF4A3 expression was correlated with low survival in the following BLCA cases: patients with a smoking history (HR = 1.55, $p = 0.013$), male patients (HR =1.59, $p = 0.012$), age <= 70 (HR =1.73, $p = 0.013$), white race (HR =1.49, $p = 0.017$), high grade (HR =1.55, $p = 0.004$), N0 stage (HR =1.76, $p = 0.018$), non-papillary subtype (HR =1.62, $p = 0.006$), T3 stage (HR =1.72,

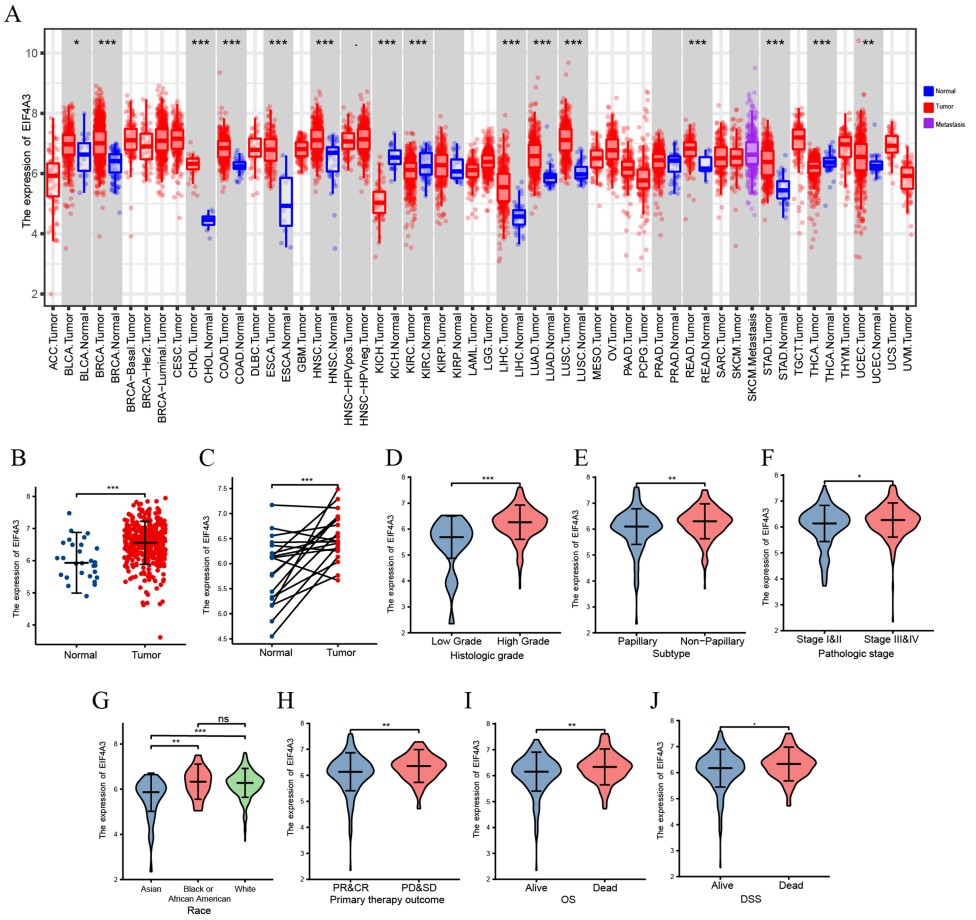

**Figure 1** **The expression of EIF4A3 in cancers.** (A) The pan-cancer expression of EIF4A3 in TCGA using TIMER2. (B) EIF4A3 expression in normal and tumor tissues in BLCA based on TCGA and GTEx datasets. (C) Investigating the differential expression of EIF4A3 in paired BLCA samples obtained from the same patient sourced from the TCGA database. The expression level of EIF4A3 in different clinical features in TCGA. ((D) histologic grade, (E) subtype, (F) pathologic stage, (G) race, (H) primary therapy outcome. (I) OS, (J) DSS). ($*p < 0.05$, $**p < 0.01$, $***p < 0.001$, ns, non-significance).

$p = 0.009$), pathologic stage III (HR $=2.04$, $p = 0.009$) (Figs. 3H–3P). BLCA patients' EIF4A3 expression was negatively correlated with the OS in univariate and multivariate cox regression analyses (HR $=1.568$, $p = 0.003$, HR $=1.698$, $p = 0.033$, respectively; Table 2). All of these results demonstrate that EIF4A3 could be an independent prognostic marker for BLCA.

## Nomogram construction based on EIF4A3 expression in BLCA

We included T, N, M, age, and pathologic stage clinical features based on univariate analysis and EIF4A3 expression to construct a nomogram for prognostic prediction in TCGA. The predicted 1, 3, and 5-year OS rates of BLCA by this nomogram were shown in Fig. S1A. In addition, the performance of the model was evaluated using ROC (receiver operating characteristic) curves and calibrations. The C-index for the nomogram was 0.665. Using the nomogram, the estimated 1, 3, and 5-year overall survival AUC values were 0.724, 0.711,

**Table 1  Correlation between EIF4A3 expression and various clinical features in BLCA based on TCGA dataset.**

| Characteristic | Low expression of EIF4A3 | High expression of EIF4A3 | p |
|---|---|---|---|
| n | 204 | 204 | |
| Gender, n (%) | | | 0.653 |
| Female | 51 (12.5%) | 56 (13.7%) | |
| Male | 153 (37.5%) | 148 (36.3%) | |
| Age, n (%) | | | 0.134 |
| <=70 | 107 (26.2%) | 123 (30.1%) | |
| >70 | 97 (23.8%) | 81 (19.9%) | |
| T stage, n (%) | | | 0.272 |
| T1 | 2 (0.5%) | 1 (0.3%) | |
| T2 | 67 (17.9%) | 52 (13.9%) | |
| T3 | 89 (23.8%) | 105 (28.1%) | |
| T4 | 27 (7.2%) | 31 (8.3%) | |
| N stage, n (%) | | | 0.296 |
| N0 | 124 (33.9%) | 113 (30.9%) | |
| N1 | 17 (4.6%) | 29 (7.9%) | |
| N2 | 37 (10.1%) | 38 (10.4%) | |
| N3 | 4 (1.1%) | 4 (1.1%) | |
| M stage, n (%) | | | 0.207 |
| M0 | 117 (56.5%) | 79 (38.2%) | |
| M1 | 4 (1.9%) | 7 (3.4%) | |
| Pathologic stage, n (%) | | | 0.125 |
| Stage I | 2 (0.5%) | 0 (0%) | |
| Stage II | 73 (18%) | 57 (14%) | |
| Stage III | 68 (16.7%) | 72 (17.7%) | |
| Stage IV | 60 (14.8%) | 74 (18.2%) | |
| Histologic grade, n (%) | | | 0.002 |
| High Grade | 185 (45.7%) | 199 (49.1%) | |
| Low Grade | 18 (4.4%) | 3 (0.7%) | |
| Subtype, n (%) | | | 0.003 |
| Non-Papillary | 122 (30.3%) | 149 (37%) | |
| Papillary | 81 (20.1%) | 51 (12.7%) | |
| Primary therapy outcome, n (%) | | | 0.024 |
| PD | 28 (8%) | 40 (11.4%) | |
| SD | 11 (3.1%) | 18 (5.1%) | |
| PR | 11 (3.1%) | 11 (3.1%) | |
| CR | 136 (38.7%) | 96 (27.4%) | |
| Race, n (%) | | | <0.001 |
| Asian | 36 (9.2%) | 8 (2%) | |
| Black or African American | 10 (2.6%) | 13 (3.3%) | |
| White | 151 (38.6%) | 173 (44.2%) | |

| Characteristic | Low expression of EIF4A3 | High expression of EIF4A3 | *p* |
|---|---|---|---|
| Lymphovascular invasion, n (%) | | | 0.934 |
| No | 63 (22.4%) | 67 (23.8%) | |
| Yes | 75 (26.7%) | 76 (27%) | |
| Smoker, n (%) | | | 0.174 |
| No | 62 (15.7%) | 47 (11.9%) | |
| Yes | 139 (35.2%) | 147 (37.2%) | |

**Notes.**

PD, progressive disease; SD, stable disease; PR, partial response; CR, complete response.

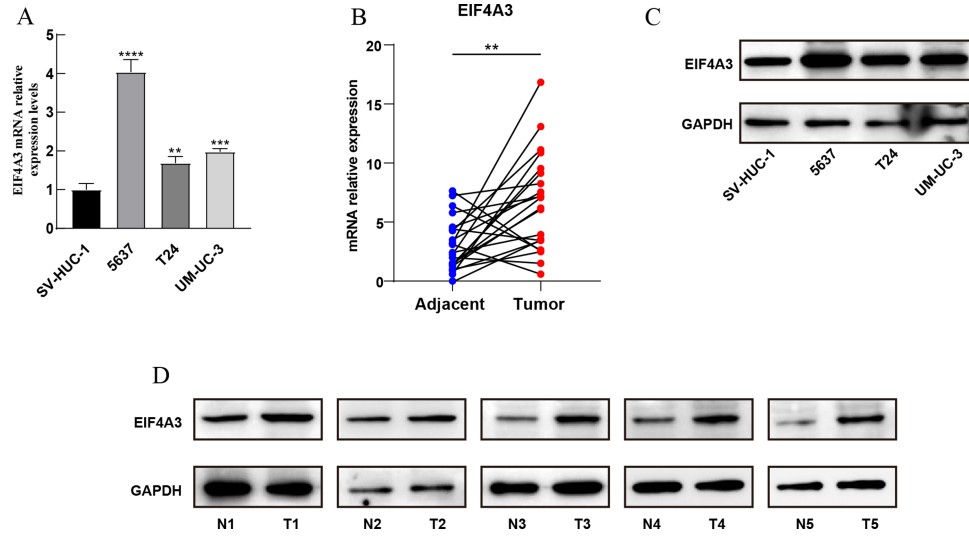

**Figure 2 Validation of EIF4A3 expression in cancer cell lines and tissue samples.** (A) The mRNA expression of EIF4A3 in bladder cell lines and (B) 22 paired clinical samples (blue dot: normal, red dot: tumor). (C) The EIF4A3 protein expression of bladder cell line and (D) paired clinical samples. (N, normal tissue; T, tumor tissue. ***p* < 0.01, ****p* < 0.001, *****p* < 0.0001).

and 0.706, respectively (Fig. S1B). The calibration plots showed that the real OS result had mostly complied with the nomogram-based predicted prognosis (Figs. S1C–S1E). These results demonstrated this model's potential clinical prognostic value.

## EIF4A3 expression is correlated with immune infiltration, PD-L1expression, and can potentially predict anti-PD-L1 therapy response in BLCA

TIMER2 analysis of the EIF4A3 expression and immune filtration coefficients revealed that immune-active cells, including CD8[+] T cells (Rho = −0.239, $p < 0.001$) and CD4[+] T cells (Rho = −0.198, $p < 0.001$) were negatively associated with EIF4A3 expression, while immune-suppressive cells, including MDSCs (Rho =0.408, $p < 0.001$), CAFs (Rho =0.12, $p < 0.05$), Treg cells (Rho = 0.165, $p < 0.01$) and M2 macrophage (Rho =0.137, $p < 0.01$) were positively correlated with EIF4A3 expression (Fig. 4A).

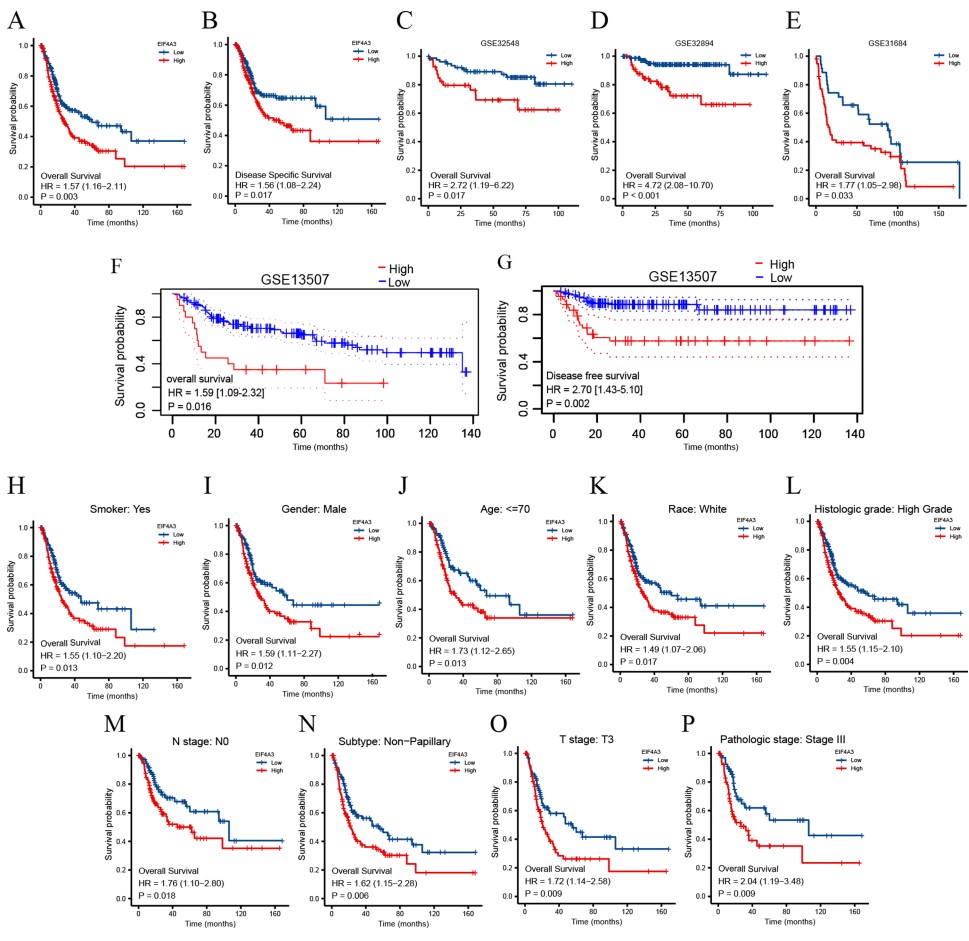

**Figure 3** Association between EIF4A3 expression and prognosis of BLCA patients in TCGA and GEO datasets. (A–B, F–G) Kaplan–Meier analysis of the OS and DSS in TCGA and GSE13507 datasets, respectively. (C–E) The OS probability of BLCA patients in GSE32548, GSE32894, and GSE31684, respectively. (H–P) Subgroup analysis of OS for smoke: yes, male, age <=70, race: white, high grade, N0, Non-papillary subtype, T3 stage, and pathologic stage III.

**Table 2** Univariate and multivariate cox regression analysis of clinical characteristics associated with prognosis in BLCA based on TCGA dataset.

| Characteristics | Total(N) | Univariate analysis | | Multivariate analysis | |
|---|---|---|---|---|---|
| | | Hazard ratio (95% CI) | P value | Hazard ratio (95% CI) | P value |
| T stage (T3/T4 vs T1/T2) | 373 | 2.092 (1.439–3.041) | <0.001 | 1.835 (0.965-3.492) | 0.064 |
| N stage (N1/N2/N3 vs N0) | 365 | 2.295 (1.676–3.142) | <0.001 | 1.895 (1.138–3.157) | 0.014 |
| M stage (M1 vs M0) | 207 | 3.310 (1.582–6.926) | 0.001 | 1.479 (0.564–3.881) | 0.426 |
| Gender (Male vs Female) | 407 | 0.874 (0.631–1.210) | 0.417 | | |
| Age (>70 vs <=70) | 407 | 1.514 (1.128–2.031) | 0.006 | 1.428 (0.879–2.320) | 0.150 |
| EIF4A3(High vs Low) | 407 | 1.568 (1.162–2.115) | 0.003 | 1.698 (1.045–2.760) | 0.028 |

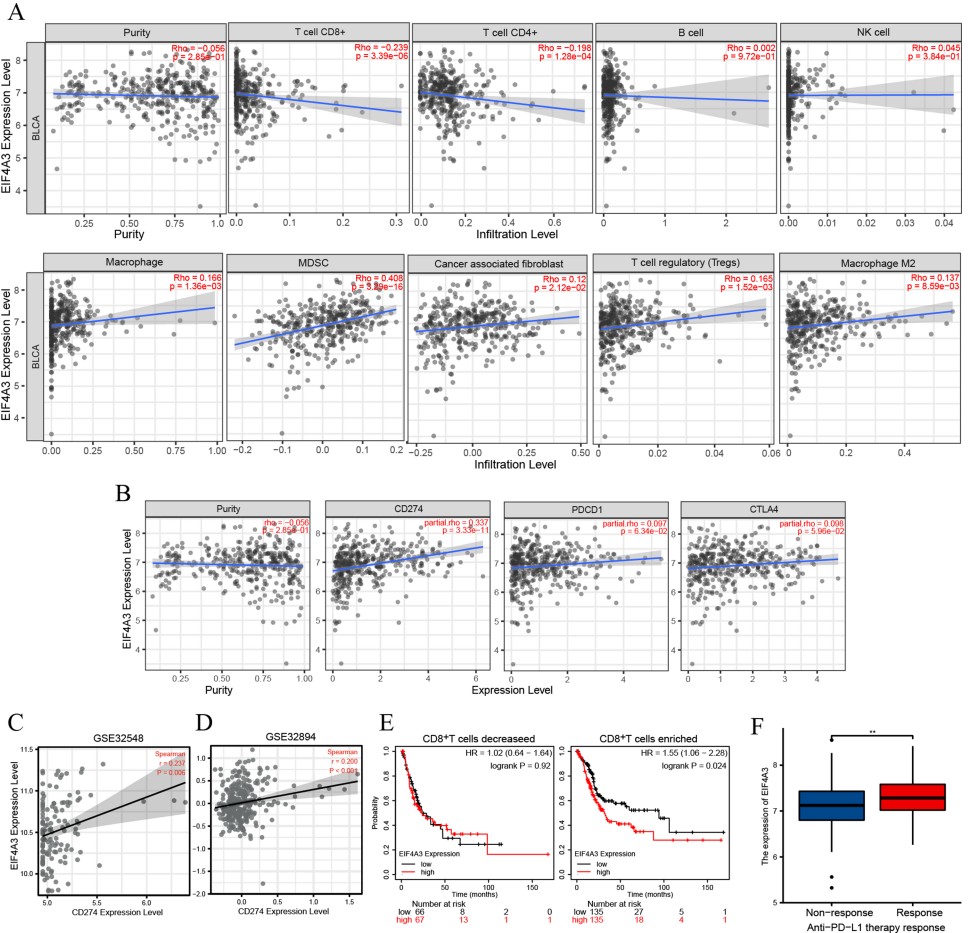

**Figure 4** **The correlation between EIF4A3 expression and immune infiltration and immune checkpoint expression.** (A) The association between EIF4A3 expression and CD8+ T cells, CD4+ T cells, B cells, NK cells, macrophage cells, MDSCs, CAFs, Treg cells, and macrophage M2. (B) The coexpressed relation between EIF4A3 and PD1, PD-L1, and CTLA4. (C–D) The coexpression relation of EIF4A3 and PD-L1 in GSE32548 and GSE32894 datasets. (E) The OS probably of BLCA patients with CD8+ T decreased or enriched. (F) The differential expression of EIF4A3 in anti-PD-L1 non-response and response groups (**$p <$ 0.01).

Checkpoint receptors such as PD1/PD-L1 and CTLA4 can be blocked to relieve CD8$^+$ T cells exhaustion and restart prime, respectively, thereby eradicating tumor cells that express antigens (*Farhood, Najafi & Mortezaee, 2019*). Thus, we analyzed the association between EIF4A3 and PD1/PD-L1 and CTLA4 expression using the TIMER2 tool. As was shown in Fig. 4B, the expression of EIF4A3 was positively correlated with PD-L1 expression (Rho $= 0.337$, $p < 0.001$), but not with PD1 and CTLA4 ($p > 0.05$) expression. Additionally, two independent external GEO datasets (GEO32548 $r = 0.237$, $p = 0.006$, GEO32894 $r = 0.200$, $p < 0.001$ respectively) confirmed the correlation between EIF4A3 and PD-L1 expression (Figs. 4C–4D). These results show that EIF4A3 is negatively correlated with CD8$^+$ T cell infiltration. Subsequently, we analyzed whether there was a difference between the OS and EIF4A3 expression in patients with enhanced or reduced CD8$^+$ T cells using

the Kaplan–Meier plotter tool. Interestingly, we found that in patients with enriched CD8$^+$ T cells, high EIF4A3 expression indicated the worst OS (HR =1.55, $p = 0.024$) (Fig. 4E), whereas a low infiltration of CD8+ T cells did not result in any differences. To evaluate the importance of EIF4A3 for clinical application in immunotherapy, we acquired the expression and therapeutic effect data of an open-source clinical study on anti-PD-L1 therapy for uroepithelial carcinoma, published in 2018 (*Mariathasan et al., 2018*). The expression of EIF4A3 was higher in patients who responded to immunotherapy compared to the non-responsive patients ($p < 0.01$) (Fig. 4F). Therefore, EIF4A3 expression was associated with the immunosuppressive microenvironment and PD-L1 expression and can thus act as a potential biomarker for immunotherapy response in BLCA patients. however, this needs to be validated by further studies.

### Gene set enrichment analysis results based on EIF4A3 expression

To identify the signaling pathways regulated by EIF4A3 overexpression, GSEA was performed by using hallmark gene sets to compare groups with high and low expression of EIF4A3 based on the TCGA dataset. The results revealed that several signaling pathways, including Hedgehog, Wnt-$\beta$-catenin, Angiogenesis, IL6-JAK-STAT3, Interferon-$\alpha$ response, PI3K-AKT-mTOR, TGF-$\beta$, and Apoptosis were regulated by EIF4A3 overexpression (Figs. S2A–S2H).

### EIF4A3 knockdown inhibited cell proliferation and promoted apoptosis *in vitro*

The above results showed that EIF4A3 is overexpressed in BLCA, implying its crucial role in BLCA tumorigenesis. GSEA analysis also suggested the elevated EIF4A3 expression was negatively correlated with the enrichment of apoptosis gene signatures. Thus, we used siRNA to knock down EIF4A3 expression to observe its biological impact on 5,637 and T24 cells, and CCK8 and cell clone formation assays were used to evaluate the effect of EIF4A3 on cell proliferation. Figs. 5A–5B suggested that the OD values and clone numbers were significantly decreased in both 5,637 and T24 cells with EIF4A3 knockdown, suggesting its proliferative role. Furthermore, cell apoptosis level analysis using flow cytometry and Annexin V and PI double staining, revealed that the percentage of apoptosis cells was significantly increased after in the EIF4A3-knockdown cell lines, compared to the control 5,637 and T24 cell lines (Figs. 5C–5D). Meanwhile, analysis of apoptosis-related biomarkers using western blots revealed that the anti-apoptotic protein, Bcl2 was significantly decreased, whereas the pro-apoptotic proteins cleaved-PARP and cleaved-caspase3 were upregulated in the EIF4A3 knockdown cells (Fig. 5E). Altogether these results demonstrate that EIF4A3 is involved in BLCA progression by promoting proliferation and inhibiting apoptosis in BLCA cell lines.

## DISCUSSION

Recently, EIF4A3 was recognized as a putative oncogene, leading to an increase in the studies determining its role in cancer development (*Kanellis et al., 2021*; *Wang et al., 2020*; *Wang et al., 2021b*; *Zheng et al., 2020*). Our results suggest that EIF4A3 is highly expressed

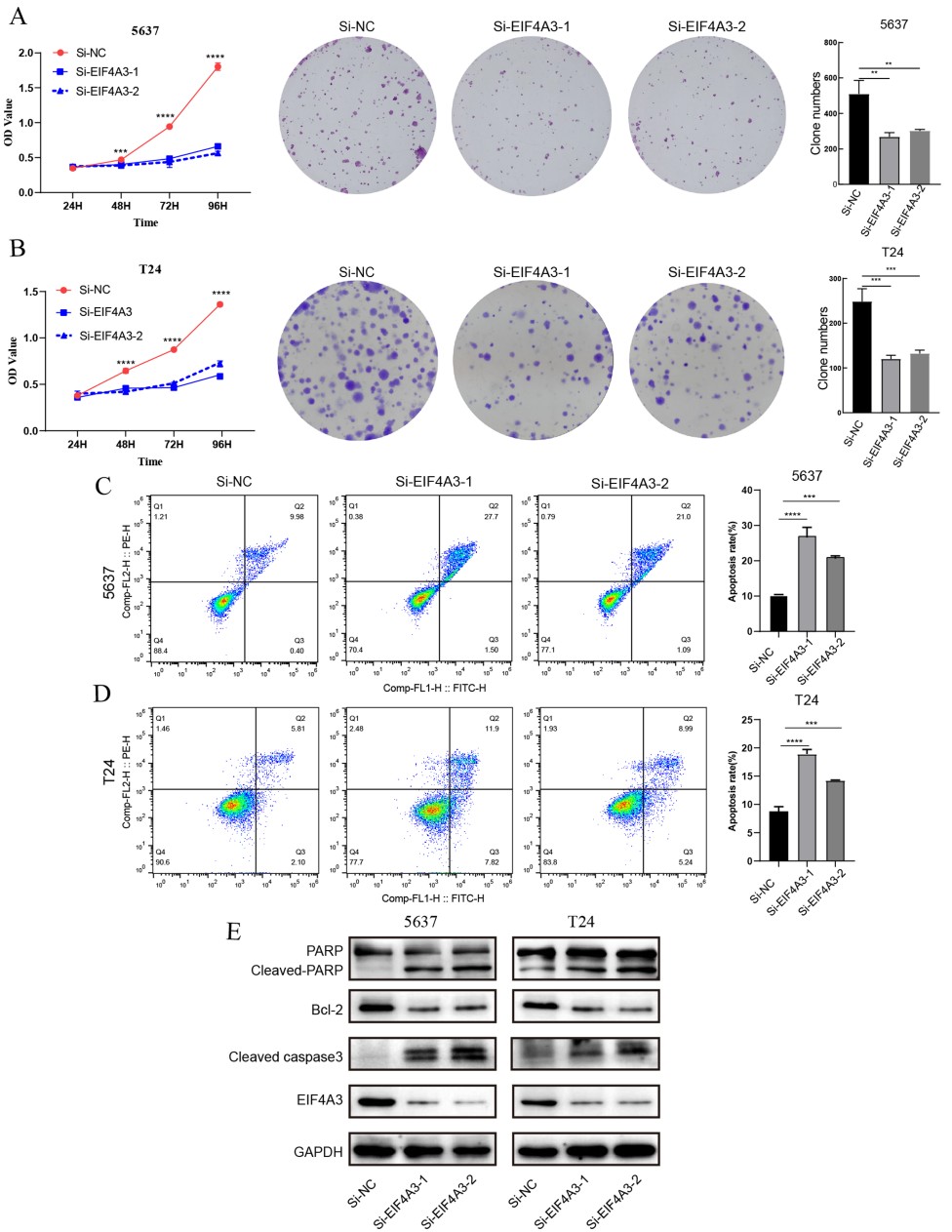

**Figure 5  The biological function of EIF4A3 in proliferation and apoptosis in bladder cancer cells.** (A–B) The impact of EIF4A3 on proliferation in 5637 and T24 cells *via* CCK8, cell clone assays. (C–D) Cell apoptosis was detected by flow cytometry in 5637 and T24 cells. (E) The detection of apoptosis-related biomarkers by western blot. (**$p < 0.01$, ***$p < 0.001$, ****$p < 0.0001$).

in most cancer types, including BLCA. *Xue et al. (2021)* reported that EIF4A3 expression is upregulated in Hepatocellular Carcinoma patients and positively correlated with the worst outcomes. While a study on gastric cancer revealed that low EIF4A3 expression is associated with poor prognosis (*Wang et al., 2021a*). Our results suggest that high

expression of EIF4A3 is correlated with poor OS and DSS in BLCA, which was further validated in four external GEO datasets.

Furthermore, our results reveal that upregulation of EIF4A3 expression may be induced under an immunosuppressive state, which is considered a hallmark of tumor progression (*Allegrezza & Conejo-Garcia, 2017*). CD8[+] T cells play a critical role in immuno-therapy and could be induced apoptosis and inhibited activation and proliferation upon the binding of PD1 (expressed on its membrane) to PD-L1 (expressed by tumor or microenvironment cells) (*Farhood, Najafi & Mortezaee, 2019*). After coming in contact with tumor cells, MDSCs derived from bone marrow tend to develop into strongly immunosuppressive PD-L1[+] macrophages (*Prima et al., 2017*). The secretion of abundant inflammatory and immunosuppressive factors is a key feature of MDSCs, which mediates the immune evasion of bladder tumor cells (*Crispen & Kusmartsev, 2020*; *Eruslanov et al., 2012*).

TAM (tumor-associated macrophage) frequently presents as M2 phenotype, leading to the formation of an immuno-suppressive microenvironment *via* the elimination of CD8[+] T cells (*Saio et al., 2001*). *Chen et al. (2021)* reported that EIF4A3-induced circRNA facilitated non-small cell lung cancer metastasis *via* mediating M2 macrophage polarization. It is also suggested that BLCA patients with increased TAM infiltration have a poor prognosis and are probably resistant to the BCG (Bacillus-Calmette–Guerin) immunotherapy (*Lima et al., 2014*; *Miyake et al., 2017*). CAFs were associated with poor prognosis and paracrine IL6 that promoted progressive phenotypes in BLCA *via* epithelial-mesenchymal transition (*Chen et al., 2020*; *Goulet et al., 2019*). Whereas, Treg cells are notably correlated with TAM and exert inhibited affection on autologous CD4[+] T cells in BLCA (*Crispen & Kusmartsev, 2020*; *Miyake et al., 2017*). GSEA results revealed a significant relationship between EIF4A3 expression and JAK/STAT3/IL6 signaling (Fig. S1D), which plays a vital role in the formation of the immunosuppressive TME (tumor microenvironment) (*Cheng et al., 2018*; *Jing et al., 2020*). Therefore, EIF4A3 could be involved in BLCA progression by reshaping the TAM.

Immune checkpoint blockade therapies, including anti-PD1/PD-L1 and anti-CTLA4, have made considerable advancements in the past decades and have become one of the primary treatments for solid tumors worldwide (*Crispen & Kusmartsev, 2020*; *Mahoney, Rennert & Freeman, 2015*). However, the immunotherapy response rate is closely related to the tumor intrinsic expression of the immune checkpoint and TAM (*Bajorin et al., 2021*; *Lopez-Beltran et al., 2021*; *Seliger & Massa, 2021*; *Sharma et al., 2020*). For instance, patients with higher PD-L1 expression had more favorable results (*Bajorin et al., 2021*). In our study on BLCA, EIF4A3 expression was significantly coexpressed with PD-L1. *Song et al. (2020)* reported that EIF4A3 regulated the PD-L1 expression *via* the NF-$\kappa$B signaling and PI3K/AKT/mTOR pathway in Hepatocellular carcinoma.

Although high EIF4A3 expression was associated with an immunosuppressive microenvironment, the results in Fig. 4F showed that patients with high EIF4A3 expression were more likely to respond to anti-PD-L1 therapy, which may be associated with the positive correlation between EIF4A3 and PD-L1 expression, consistent with the results reported in previous literature. In summary, high expression of EIF4A3 may contribute to tumor progression by remodeling the immune microenvironment in BLCA patients;

however, owing to its association with PD-L1 expression, it may serve as a predictor for immunotherapy response.

Few studies have determined the direct effect of EIF4A3 on cancer development. Studies by *Mao et al. (2021)* and *Fu et al. (2021)* suggested that silencing of EIF4A3 can significantly inhibit prostate cancer cell proliferation, migration, invasion, and epithelial-mesenchymal transition. In our study, we demonstrated that knockdown of EIF4A3 could inhibit proliferation and induce apoptosis in BLCA cells and the underlying mechanism could be correlated with the activation of the PI3K/AKT/mTOR pathway (Fig. S1F) which is considered a key modulator to inhibit apoptosis and regulate cell survival (*Fresno Vara et al., 2004*). Meanwhile, the selective EIF4A3 inhibitor eIF4A3-IN-2 decreased osteoblastogenesis and inhibited the bone metastasis induced by breast cancer cells (*Xu et al., 2021*). Additionally, the regulation of the long non-coding RNA CASC2-EIF4A3 axis by sanguinarine significantly inhibited the progression of epithelial ovarian cancer cells (*Zhang et al., 2018*). Based on these findings, we speculate that EIF4A3 could be a potential therapeutic target.

This study has several strengths. Firstly, we identify that EIF4A3 expression is upregulated and correlates with poor prognosis. Secondly, high EIF4A3 expression is associated with the immunosuppressive tumor microenvironment and predicts a high anti-PD-L1 therapy response rate. Finally, we prove that high EIF4A3 expression promotes proliferation and inhibits cell apoptosis *in vitro* in BLCA. However, there are some limitations to our study. Our sample size was small; therefore the prognostic value of EIF4A3 should be validated in more clinical samples. Furthermore, the specific mechanisms of how EIF4A3 regulates apoptosis and reshapes the immune microenvironment in BLCA *in vitro* and *in vivo*, and the impact of specific EIF4A3 inhibitors on the efficacy of immunotherapy need to be further investigated in depth. These will be the focus of our future research. By addressing the above issues, we anticipate combining EIF4A3 inhibitors with immune checkpoint inhibitors to improve the prognosis of BLCA patients in the future.

## CONCLUSION

In summary, we found that EIF4A3 expression increased in BLCA and was associated with poor prognosis. Further, we depicted that EIF4A3 expression relates to an immunosuppressive TME by upregulating the immune-suppressive cells and downregulating the immune-active cells. Moreover, high EIF4A3 expression patients also overexpressed PD-L1 and were more likely to respond to anti-PD-L1 therapy. Lastly, we found that apoptosis could be one of the mechanisms involved in the pro-carcinogenic role of EIF4A3.

## ACKNOWLEDGEMENTS

We thank Bullet Edits Limited for the linguistic editing and proofreading of the manuscript.

### Funding

This study was funded by the Jiangxi Province "Double Thousand Plan" the first batch of science and technology innovation high-end talent (No. jxsq2019201027), and the Key Project of Natural Science Foundation of Jiangxi Province (No. 20212ACB206013). The funders had no role in study design, data collection and analysis, decision to publish, or preparation of the manuscript.

### Grant Disclosures

The following grant information was disclosed by the authors:
Jiangxi Province "Double Thousand Plan" the first batch of science and technology innovation high-end talent: jxsq2019201027.
Key Project of Natural Science Foundation of Jiangxi Province: 20212ACB206013.

### Competing Interests

There are no competing interests between the authors.

### Author Contributions

- Bing Hu conceived and designed the experiments, performed the experiments, analyzed the data, prepared figures and/or tables, authored or reviewed drafts of the article, and approved the final draft.
- Ru Chen performed the experiments, prepared figures and/or tables, and approved the final draft.
- Ming Jiang performed the experiments, analyzed the data, prepared figures and/or tables, and approved the final draft.
- Situ Xiong analyzed the data, prepared figures and/or tables, and approved the final draft.
- Xiaoqiang Liu conceived and designed the experiments, authored or reviewed drafts of the article, and approved the final draft.
- Bin Fu conceived and designed the experiments, authored or reviewed drafts of the article, and approved the final draft.

### Human Ethics

The following information was supplied relating to ethical approvals (*i.e.*, approving body and any reference numbers):
The Ethics Committee of the First Affiliated Hospital of Nanchang University provided approval to carry out the study within its facilities.

### Data Availability

The raw measurements and clinical data of public datasets are available in the Supplementary Files.
The raw data from the public datasets are available at TCGA-BLCA and NCBI GEO: GSE32894, GSE32548, GSE31684, and GSE13507.

## Supplemental Information

Supplemental information for this article can be found online at http://dx.doi.org/10.7717/peerj.15309#supplemental-information.

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
