# Peer review of "EIF4A3 serves as a prognostic and immunosuppressive microenvironment factor and inhibits cell apoptosis in bladder cancer"

_PeerJ, doi:10.7717/peerj.15309_

## Round 0.1 · original submission · Minor Revisions

The manuscript is well-written and organized. I appreciate the authors for describing the strengths and limitations of the study. However, there are minor concerns from the reviewers that need to be addressed before being considered for publication. The grammatical and syntax errors need to be fixed throughout the manuscript. The authors are requested to elaborate on the methodology section. Figure labelling and legends need careful revision.

Reviewer 1 ·

Basic reporting

The manuscript is well written, and message is very clear.

Experimental design

Experiments are designed and performed well.

Validity of the findings

The findings can improve bladder cancer management.

Annotated reviews are not available for download in order to protect the identity of reviewers who chose to remain anonymous.

Reviewer 2 ·

Basic reporting

In the present manuscript, using various bioinformatic approaches and in vitro assays, authors have demonstrated that the high expression of EIF4A3 is associated with poor prognosis and immunosuppressive microenvironment in bladder cancer (BLCA). Using in vitro assays, they have further shown that high EIF4A3 expression promotes proliferation and inhibits cell apoptosis in BLCA.
The theme of this investigation is interesting and the manuscript is well written. The introduction section clearly outlines the importance and need of the present study. Overall, the methods employed in the study are described in sufficient details. The data are presented in a logical manner with suitable statistics and are appropriately discussed with sufficient reasoning. The strengths and limitations of the study have been well described. However, there are a few concerns that need to be addressed:
• Authors need to include control (non-cancerous) cell line in addition to the bladder cancer cell lines for in vitro assays (proliferation, migration and apoptosis assays after EIF4A3 knock-down).
• The manuscript should be read carefully to correct minor grammatical errors, punctuation errors and labeling in figure panels.
• PCR cycling conditions for qRT-PCR should be provided.
• Which secondary antibodies were used for western blot assays?

Experimental design

__

Validity of the findings

__

Additional comments

__

Reviewer 3 ·

Basic reporting

1. The title can be modified to be brief and clear.
2. The following sentence can be direct: “There were83730 estimated new cases and 17200 estimated deaths of urinary BLCA, respectively, in the United States according to the cancer statistics of 2021.”
3. Please correct spelling errors like somking instead of “smoking history”.
4. Modify the caption of Figure 2 to “Validation of EIF4A3 expression in cancer cell line and tissue samples.

Experimental design

The design and the aim of the work is clear on the expression of EIF4A3,
The methods and tools sufficiently support the study.

5. Why have the authors used univariate analysis instead of multivariate analysis?

Validity of the findings

no comment

Additional comments

The authors have focused on the expression of EIF4A3 associated with poor prognosis in bladder cancer. Several checkpoints and associated expression factors like CD 8, PD-L1 have been assessed and analyzed using statistical tools to understand their correlation. The potential of this factor to be used as a biomarker has also been found with extensive studies. In-vitro studies have also substantiated the results using cancer cell lines.

·

Basic reporting

In this manuscript by Hu et al., the authors have studied the expression and oncogenic functional effect of the EIF4A3 (Eukaryotic translation initiation factor 4A3) in bladder cancer primarily through bioinformatic tools, western blot and qPCR analysis. They have found EIF4A3 to be highly expressed in bladder cancer and correlates with tumor grade, type and race. High expression of EIF4A3 was associated with worse survival as apparent from TCGA and other GEO databases. Further, EIF4A3 negatively correlated with immune checkpoint blockers PD-L1 and CTLA4 and KD of EIF4A3, enhanced apoptotic cell death in T24 and 5637 bladder cancer cells.
Data has been presented well and statistics are satisfactorily performed. Some major and minor comments are mentioned below.

1. In Figure 1A, the authors should provide a color coding key for the red and blue colors.

2. In Figure 1C, do the paired samples represent cancer tissue and adjacent bladder tissue from the same patient? Please specify in the legend.

3. The authors can move the text explanation for Figures 1D-J before Figure 2A, so as to maintain a continuity in the results section.

4. In Figure 4E, the authors should discuss the worse survival of patients with high EIF4A3 expression and CD8+T cell enrichment. In Figure 4A it is shown that EIF4A3 expression is negatively correlated to CD8+T infiltration. How does infiltration of CD8+T cells decreases survival in these patients?

5. In lines 330-331, the authors should reword ‘correlated’ to ‘negatively correlated’.

6. In Figure 5, does KD of EIF4A3 in the 5637 and T24 cell lines, increase their T cell interaction in a co-culture experiment?

Experimental design

Please see above comments

Validity of the findings

Please see above comments

Additional comments

Please see above comments

---

## Round 0.2 · Minor Revisions

The authors have adequately addressed all the comments. The source of normal tissues in the paired samples has to be provided in the figure (1C) legend before being considered for publication.

Reviewer 2 ·

Basic reporting

The inclusion of non-cancerous cell line is to demonstrate that the impact of EIF4A3 knock-down is limited to cancer cells exclusively, thereby validating that the observed effects are unique to bladder cancer cells. Although this approach will not yield further understanding of the particular molecular mechanisms under study, it does establish specificity in the findings. With the exception of this aspect, the authors have adequately addressed other issues raised.

Experimental design

_

Validity of the findings

_

Additional comments

_

·

Basic reporting

The authors have satisfactorily answered all questions. I just have one minor comment.

1. In response to comment 2, the authors should clearly specify the source of the normal and tumor tissue from the same patient in the legend.

Experimental design

Satisfactory

Validity of the findings

Satisfactory

Additional comments

No additional comments

---

## Round 0.3 · accepted · Accept

The authors have addressed all the comments satisfactorily and the manuscript is ready for publication.